# Emulsion Nanofibres as a Composite for a Textile Touch Sensor

**DOI:** 10.3390/polym15193903

**Published:** 2023-09-27

**Authors:** David Mínguez-García, Pablo Díaz-García, Jaime Gisbert-Payá, Marilés Bonet-Aracil

**Affiliations:** Departamento de Ingeniería Textil y Papelera, Universitat Politècnica de València, 03801 Alcoy, Spain; damingar@epsa.upv.es (D.M.-G.); pdiazga@txp.upv.es (P.D.-G.); jaigispa@txp.upv.es (J.G.-P.)

**Keywords:** electrospun, polyvinyl alcohol, fibre, colour

## Abstract

The combination of a nanofibre net and textile support represents an interesting composite capable of conferring various properties. Nanofibres are so thin that they can be easily damaged by human touch. In this study, we hypothesised that dyeing nanofibres with different colours from their textile supports would result in a colour difference upon their degradation, providing evidence that the composite has been touched and acting as a touch sensor. Two different methods were studied: directly inserting the dye into the polymer via electrospinning or creating a coloured liquid emulsion encapsulated by the polymer via electrospinning. Two black dyes were studied. Colour index (CI) Acid Black 194 was added directly to polyvinyl alcohol (PVA) as the polymer. Sage oil was used for CI Solvent Black 3. The nanofibre nets were conveniently electrospun on a white polyester fabric; the fabrics were then characterised by colour coordinate analysis, FTIR, and SEM. The results showed that the dyed solution in oil was encapsulated, and the black colour could only be observed when rubbed, whereas the dyed polymer showed a black colour that was removed when rubbed. Therefore, the hypothesis was confirmed, and both samples demonstrated the desired touch sensor behaviour.

## 1. Introduction

The application of nanofibrous veils in multiple industrial sectors has increased dramatically in recent decades. Interest in electrospun nanofibres has arisen due to their exceptional characteristics, high surface-area-to-volume ratio, high porosity, ultra-thin interconnected and lightweight fibrous structure, and their potential for use with a multitude of synthetic and natural polymers and compounds that provide different characteristics [1]. Nanofibre veils have been widely studied in filtration-related fields, including air filtration [2,3,4], water filtration [5,6,7], biomedicine [8,9,10], sensors [11,12], soundproofing [13,14], defence and protection garments [15,16], food packaging [17,18], cosmetics [19,20], and home furnishing [21,22].

Currently, across various fields of application, there is a growing need for the creation of a singular matrix that possesses multiple functional traits. By integrating functional nanoparticles and active ingredients into electrospun fibres, it becomes possible to fulfil the demands for enhanced mechanical strength, electrochemical properties, and other technical performances all at once. In the food sector, authors have combined thymol essential oil with PLGA nanofibres via coaxial electrospinning to prevent microbial growth in fruits during storage [23]. In the biomedicine field, the addition of camptothecin (CPT), a natural hydrophobic chemotherapy agent, to electrospun amphiphilic peptide (AP) nanofibres has demonstrated the inhibition of breast cancer tumour growth [24]. Nanofibre veils have also been used in the design of protective clothing. The activation of Ag nanoparticles and Zn nanowires in electrospun poly(methyl methacrylate) (PMMA) nanofibres has provided conventional garments with antibacterial, antiviral, and self-cleaning properties [25].

To date, most studies related to nanofibres have mainly focused on their functional properties and technical applications, leaving the aesthetic aspect of these nano-sized structures largely unexplored. While significant progress has been made in understanding and exploiting the unique properties of nanofibres and the other technical characteristics of various materials, it is surprising to note that the visual and aesthetic impact of these fibres has often taken a back seat. As nanofibres are increasingly integrated into various products and materials, it is essential to consider how they will visually affect the result. It is essential to recognise that aesthetics is not merely a superficial aspect but can influence the overall perception of a product’s quality and usefulness.

The study of the aesthetic aspect of nanofibres involves understanding how they interact with light, how they influence colour perception, and how they can be modified to achieve desirable visual effects [26]. Regarding the interplay between light and materials, a single nanofibre is not visible to the naked eye or through an optical microscope because the obtained nanometer diameters (50–1000 nm) [27] are generally in the same wavelength magnitude range as visible light (400–700 nm) [28]. However, a single nanofibre is only visible with an electron microscope. While it is feasible to visually detect the existence of nanofibres through the naked eye or optical microscopy under circumstances where these nanofibres create a dense layer enveloping a substrate, the reality is that a substantial accumulation of nanofibres, when illuminated by white light, presents itself as a white, impenetrable, and uniformly even surface. This white colour occurs when a thick layer of nanofibres is produced due to the physical phenomenon of light scattering [26].

Recent advancements in nanofibre dyeing have enabled the enhancement of their aesthetic properties. Researchers have studied two methods of dyeing nanofibres. The first method involves the incorporation of the dye compound directly into the polymer solution before the electrospinning process [26,29,30]. As the fibres are generated from the solution and deposited on the collector structure, an increase in the deposition of nanofibres leads to an increase in the perceived colour intensity. This approach can achieve more intense and uniform colours from the outset due to the direct incorporation of dyes into the fibre structure itself. In contrast, the other method of dyeing nanofibres takes place after the electrospinning process, i.e., the fibres are immersed in a dye solution [31,32,33]. This allows more precise control over the intensity and uniformity of the colour, as the fibres can be immersed in the solution for different periods to achieve the desired shade.

In this work, we focused on composites made of a textile and coated with a nanofibre layer. The nanofibres were colour-treated by encapsulating dyes inside them using two different techniques: polymer colouration and liquid solution encapsulation. Two different polymer solutions were prepared for electrospinning, thus comprising two different types of dyeing. To obtain one solution, the dye was dispersed directly on an aqueous polymeric solution of polyvinyl alcohol (PVA). To prepare the other solution, the dye was solubilised in an oily solution of sage essential oil, which was then emulsified in a polymeric solution of PVA with a surfactant. PVA was selected as the polymer because it is a water-soluble polymer and is compatible with electrospinning. The two dyes were used, one water-soluble and the other oil-soluble, and were black in colour. This colour was selected because of its high contrast with the surface where the nanofibres were collected (which was white), as, in this way, the colour can be seen with the naked eye and with short electrospinning process times. The dyes were selected based on their molecular weights since their very similar values rule out their influence on the dyeing power of the nanofibres or the obtained colour. Sage essential oil was used because it has shown good levels of encapsulation in the nanofibres in previous studies as well as because it is an almost colourless oil, which facilitates the appreciation of the dyeing.

## 2. Materials and Methods

### 2.1. Materials

Polyvinyl alcohol (PVA) with a molecular weight (Mw) of 61,000 g/mol was obtained from Sigma-Aldrich (Akralab, Alicante, Spain) and used to create the solutions. Distilled water was utilised to prepare both solutions. As colouring compounds, the water-soluble dye Acid Black 194 (AB194) supplied by BASF (Tarragona, Spain) and the oil-soluble dye Solvent Black 3 (SB3) supplied by ANALEMA were used, its chemical structure is shown in Figure 1. The essential oil used in the experiment was sage, which was purchased from Lozano Essences (Esencias Lozano, S.L., Murcia, Spain). For surfactant purposes, Tween 80 from Panreac (Akralab, Alicante, Spain) was employed.

The nanofibres were deposited on a plain white weave fabric comprising 100% polyester (PES), 200 g/m^2^.

### 2.2. Methods

#### 2.2.1. Preparation of the Solutions

Both solutions were obtained by preparing a 9% (*w*/*v*) PVA solution. This involved heating the water to 80 °C via magnetic stirring until the PVA was completely dissolved. To prepare the solution with the Acid Black 194 dye, it was added to the PVA solution at a concentration of 5 g/L and left to stir for 2 h at room temperature in a propeller homogeniser. To prepare the emulsion, the essential oil was added to the PVA at room temperature. Sage oil was added at 3% (*v*/*v*) and Tween 80 surfactant at 1% (*v*/*v*) to the PVA solution. Initially, the PVA solution was placed in the homogeniser and when the desired speed was reached, the amount of Tween 80 was gradually added. Subsequently, a set amount of sage essential oil was gradually added. After the addition of oil was finished, the revolutions were sustained for 5 min. The homogeniser and its respective conditions were as follows: an Ultraturrax homogeniser with a toothed accessory at 4000 rpm for 10 min for the surfactant and 1500 rpm for 5 min for the sage oil.

#### 2.2.2. Characterisation of the Solutions

The obtained solutions were characterised using a variety of methods. The viscosity of the two solutions was measured with the Visco Elite R viscometer (Fungilab, Sant Feliu de Llobregat, Spain). The selection of the measuring device was based on the manufacturer’s guidelines for the specific measuring range achieved. For each solution, the conductivity was measured using a Crison Conductimeter Basic 30 (Hach Lange Spain, S.L.U., L’ Hospitalet de Llobregat, Spain). Additionally, the surface tension was measured using a Krüss tensiometer K9 (Krüss, Hamburg, Germany). The pH was measured with the GLP 22 CRISON pH metre (CRISON, Barcelona, Spain). Three samples were prepared, and every sample was tested for each parameter (viscosity, conductivity, surface tension, and pH) five times.

#### 2.2.3. Electrospinning Process

The electrospinning process was conducted using a BIOINICIA electrospinning system (Bioinicia, Paterna, Spain). A 100% polyester (PES) bleached fabric with a taffeta weave was placed on the vertical stainless steel flat collector, where the nanofibres were later collected. A 22-gauge extruder capillary was used for all electrospinning processes. The parameters of the electrospinning process are listed in Table 1, which were optimised to avoid beds [34].

The parameters given in Table 1 were selected to facilitate a constant and stable electrospinning process. A balance between the selected voltage and the flow rate is imperative; if an excessively low flow rate is applied relative to the applied voltage, electrospinning will be interrupted, whereas if a greater flow rate than the voltage can withstand is applied, an irregular Taylor cone will form and solution droplets will be deposited on the collector, not just the nanofibres. The nozzle–collector distance was selected so that the solvent used in the solutions evaporates in that space–time. All parameters have proven effective in previous research.

#### 2.2.4. Characterisation of Nanofibres

The first characterisation of the nanofibres was performed to determine if there was visible colouring to the naked eye due to the encapsulation of the dye in each of them. Three different electrospun samples, PVA nanofibres, PVA + Acid Black 194 nanofibres, and PVA + sage oil + Solvent Black 3 nanofibres, were presented to five individuals who were asked to determine the appearance of colour in each sample referenced with letters of the alphabet, avoiding the power of suggestion on the volunteers’ opinions.

Subsequently, in parallel with the opinions of the five individuals, reflection spectroscopy was carried out to determine the chromatic coordinates of the produced nanofibres. For this purpose, measurements were carried out with the Datacolor Spectro 700 reflection spectrophotometer; the measurement parameters were illuminant D65 and observer at 10°, including the specular component, and an aperture size of 3 cm. Analysis of the samples’ colours was conducted via the CIEL*a*b* colour space, as it is the most comprehensive colour space. The three coordinates of CIELAB space represent the lightness of the colour (L* = 0 indicates black and L* = 100 indicates white), its position between red and green (a* < 0 indicates green colour while a* > 0 indicates red), and its position between yellow and blue (b < 0 indicates blue while b < 0 indicates yellow colour). The three veils were measured, and each one was measured in five different zones.

For the morphological characterisation, scanning electron microscopy (SEM) was employed via a FESEM ULTRA 55 (Carl Zeiss, Jena, Germany) with an accelerating voltage of 2 kV. The surfaces of each sample were analysed at appropriate magnifications for the study. Before observation, the samples were coated with gold/platinum to ensure the necessary conductivity.

Fourier transform infrared spectroscopy (FTIR) was conducted to characterise the starting materials (PVA, dyes, and sage oil) and the obtained nanofibres. A JASCO FT/IR-4700 type A spectrophotometer (Jasco Spain, Madrid, Spain) with an ATR accessory (Jasco Spain, Madrid, Spain) was used. Each sample underwent sixteen spectra measurements with a resolution of 4 cm^−1^.

To ensure uniformity in size distribution measurements, we utilised the image analysis software Image J 1.52p (Wayne Rasband, MD, USA). Each image of the samples under analysis was accurately calibrated to obtain measurements in the correct units. The obtained measurements were then transferred to Excel, and corresponding representations were generated. Three different nanofibre veils were obtained, and 20 measures were made from every image.

Finally, to evaluate the staining capacity of the nanofibres after subjection to a 20-cycle rubbing process, the Crockmeter model CM-1 from ATLAS ELECTRIC DEVICES Co. (Chicago, IL, USA) was used. The control fabric used during the test was the same fabric on which the nanofibres were deposited, i.e., a 100% polyester fabric.

## 3. Results

### 3.1. Solution Characterisation

The characteristics of all the solutions and the components used are given in Table 2. All measurements were carried out at room temperature so as not to influence the parameters.

We correctly characterised the solutions by measuring the viscosity, surface tension, conductivity, and pH of each one, as well as the starting solution of 9% (*w*/*v*) PVA. We also measured the solution parameters of Solvent Black 3 dye and sage oil to determine their influence on the emulsion. Table 2 shows how the viscosity values increase with the addition of the colouring substances in the PVA solution. However, the opposite was true for surface tension values, which decreased. The conductivity parameter increased considerably when AB194 dye was added; however, it decreased slightly when SB3 dye and sage oil were added. The pH values remained partially stable.

Figure 2 shows the images of the solutions under an optical microscope. As expected, no compounds or aggregates were found in the PVA solution (Figure 2a), which means that the polymer was correctly dissolved. The same happened with the dissolution of the AB194 dye in the PVA solution (Figure 2b); a completely homogeneous solution was observed due to the great solubility of the dye in the aqueous solution. However, at first glance, it can be observed that the solution had lost its initial transparency attributed to the PVA and now had a high black colouring. Finally, due to the hydrophilic and hydrophobic nature of the components, an emulsion was obtained between PVA (aqueous) and sage essential oil dyed with Solvent Black 3 (oil). Figure 2c shows the microdroplets generated by the emulsion, which were characterised by a homogeneous distribution of the encapsulations’ radial sizes.

Maintaining the stability of an emulsion over time is complicated, and this becomes even more difficult when voltage is applied to the solution. The electrospinning technique requires the creation of an electrostatic field between two electrodes by applying an electric current to the extruder component. Figure 3 shows the effect of voltage application on the stability of the emulsion.

The application of electric current to an emulsion can separate its component phases; in this case, PVA is the aqueous phase and sage oil is the oily phase. For years, many researchers have shown interest in electrostatic phase separation for the separation of oil from water. When an electric current is applied to an emulsion, electrocoalescence can occur. This means that when an electric field is applied, the dispersed emulsion droplets start to move and experience electrostatic forces that can lead to coalescence until they merge. In addition, if the outer membrane of the droplet becomes too narrow, it can break and separate the dispersed and dispersing phases [35,36]. Figure 3a shows a big difference from the initial emulsion image; the encapsulated droplets were greatly reduced, and a different size distribution was generated. Figure 3a shows a big difference concerning the image of the initial emulsion. After applying electric current in the electrospinning process, the encapsulated droplets were greatly reduced, and a different size distribution was generated. In addition, it can be seen (Figure 3a, marked in red circles) that the encapsulation of the oil in the droplets has apparently decreased. However, as shown in Figure 3b, the emulsion that was retained in the syringe of the equipment was stable, just like the initial emulsion, and the encapsulated black-dyed sage oil could be seen (see Figure 4).

### 3.2. Nanofibre Characterisation

#### 3.2.1. Visual Colour Test

The visual staining test was carried out with five volunteers who were shown three samples and asked to answer which one had black staining. Figure 5 shows the electrospun dye nanofibres.

As can be seen from the survey results (Table 3), all respondents were able to identify the colour black in only one sample; the other two apparently showed a white colour.

#### 3.2.2. Diffuse Reflectance Spectroscopy

Diffuse reflectance spectroscopy was used to perform a detailed and quantitative analysis of the obtained colour data. The chromatic coordinates of the samples were determined by averaging at least four measurements of the samples.

Table 4 shows the CIE Lab* colourimetry values obtained for each sample. The 100% PES textile was used as the standard sample because the nanofibres were electrospun on it.

Figure 6 shows the location of the values obtained in CIE Lab* space. The values obtained for the PVA nanofibres (yellow point) showed minimal colour variation with respect to the white polyester fabric. This could be because, although the PVA solution is transparent in its liquid state, when it solidifies, it turns white or slightly yellow. The brightness of the sample was very similar to the standard sample.

On the contrary, the results showed that the addition of Acid Black 194 dye to the PVA solution generated bluish-coloured nanofibres, as indicated by its value (−1.73 b*), which can be seen in the position of the orange point in Figure 6b. This result is compatible with what the five volunteers answered in the visual test. It also decreased the brightness value (76.39 L*) of the sample compared to the standard (95.29 L*) due to dyeing.

However, although the emulsion of PVA + sage oil + black solvent 3 had a greyish colour in the liquid state, this colour did not appear on the nanofibres when they were electrospun. The CIE Lab* measurements showed a value of −0.88 for parameter a* and a value of 2.07 for b* for the nanofibres generated from the PVA + sage oil + Solvent Black 3 emulsion. These values are very close to those obtained for the standard sample (−0.89 a* and 1.91 b*), which indicate that the nanofibres generated from the emulsion did not show a black colouring on their surface but showed a slightly yellowish–whitish shade, similar to the PVA nanofibres.

#### 3.2.3. Rubbing Test

We sought to determine what happens to both types of nanofibres when subjected to a rubbing process. After the test, the colours of the rubbed sample and the control used for the rubbing were evaluated. Table 5 shows the colourimetry results obtained after rubbing.

Figure 7 shows the brightness results obtained after the rubbing tests. Concerning the nanofibres containing the AB194 dye (PVA + Acid Black 194), it can be seen that after performing the test on the electrospun sample (PVA + Acid Black 194_Rubbed), the brightness value of the latter increased to 91.45 L*, a value that almost reached that of the standard PES sample, 95.29 L*. In the same way, it can be seen that the brightness value of the control fabric (PVA + Acid Black 194_Control), which would initially be 95.29 L* as it is the same standard fabric, decreased to 76.91 L*, a value very similar to that of the PVA + Acid Black 194 sample, 76.39 L*. These results can be explained by the fact that the rubbing process caused the nanofibres to detach from the PES fabric, where they were deposited and transferred to the used control fabric.

However, when analysing the brightness values of the rubbed sample with the encapsulated SB3 dye (PVA + sage oil + Solvent Black 3_Rubbed) and the control fabric (PVA + swage oil + Solvent Black 3_Control), these values are not very different from those obtained with the original sample.

Figure 8 shows the colour difference results obtained after the rubbing tests.

The colour difference data obtained corroborate the abovementioned; in the case of the PVA + Acid Black 194 sample, the dyed nanofibres or dye was directly transferred to the control fabric (PVA + Acid Black 194_Control) used. For this reason, the control, which was initially white after the test, had a DE*ab value of 18.95. As the value increased in the control sample, it decreased in the rubbed sample, as it obtained a value of 4.01 DE*ab.

After the rubbing of the PVA + sage oil + Solvent Black 3 sample, the DE*ab value increased slightly (1.07 DE*ab) compared to the original sample (0.34 DE*ab). However, a higher increase in the control sample was observed, which reached a value of 1.89 DE*ab.

Despite the rubbing tests, a greater colour difference was still observed in the samples containing the Acid Black 194 dye compared with the nanofibres produced from the emulsion with sage oil and Solvent Black 3 dye.

#### 3.2.4. FTIR

Fourier transform infrared spectroscopy (FTIR) is highly valuable in discerning functional groups exhibiting distinct vibrations within specific spectral ranges, typically between 4000 and 400 cm^−1^. Nonetheless, achieving quantitative determinations demands meticulous calibrations and involves increased complexity due to the overlapping molecular vibrations in certain regions of the spectrum [37]. These overlaps can lead to fluctuations in the centre of the vibrational bands, making accurate measurements more challenging. In this case, we intended to use FTIR to identify the presence of the different dyes and the essential oil of sage in the nanofibres so as to identify the functional groups characteristic of the type of dye and the essential oil used and determine the evolution of the PVA curve when the additive compounds are included.

When analysing the spectrum of the nanofibres obtained from the PVA + Acid Black 194 solution and comparing it with the spectra of the PVA and the dye, the presence of peaks characteristic of both can be seen in Figure 9. This graphic shows the polyvinyl alcohol nanofibre in blue, the spectrum of the AB194 dye in orange, and the spectrum of the PVA + Acid Black 194 in grey.

The characteristic peaks, named A, B, and C, have already been cited by several authors who have analysed the same or similar compounds. Point A marks a characteristic peak of PVA at wavelength 3300 cm^−1^; this is due to its -OH groups [38]. The characteristic peak of the dye called B at 1580 cm^−1^ is due to the presence of the azo bond (-N=N-), characteristic of azo dyes [39,40]. Finally, the C peak at wavelength 1100 cm^−1^ shows the aromatic nature of the dye, i.e., it reflects the presence of carbon–carbon bonds (C=C) [40,41].

The presence of the AB194 dye should provide an increase in the -N=N- band with respect to the OH band. This is reflected in Table 6, where the spectrum of PVA nanofibres, the spectrum of Acid Black 194, and the spectrum of the PVA+ Acid Black 194 nanofibre veil are analysed.

The intensity analysis showed that the presence of AB194 in the PVA nanofibres increased the ratio (I_1580_/I_3300_ = 1.3516) with respect to that of the nanofibres without dye (I_1580_/I_3300_ = 0.1650). The opposite occurred when the C=C band was studied with respect to the OH band. In this case, the presence of the dye could be seen when the absorbance of the coloured nanofibres (I_3300_/I_3100_ = 0.4684) decreased with respect to the PVA nanofibres (I_3300_/I_3100_ = 0.7946). In this way, it is possible to demonstrate the presence of Acid Black 194 in the PVA nanofibres, even if it is already visually observed.

Similarly, to determine the presence of sage oil and SB3 dye in the electrospun PVA nanofibres, their FTIR spectra were analysed and are shown in Figure 9b. In this case, the remarkable peaks were named D, E, F, G, and H.

The peak centred around point D aligned with the peaks at 3300 cm^−1^, primarily indicating OH stretching. This peak was notably absent in sage oil. Adjacent to this feature, we observed another peak labelled E, aligning with the range of 2956–2849 cm^−1^. This range is attributed to both symmetrical and asymmetrical stretching of CH bonds (in CH3 and CH2 groups) [42]. Notably, this band exhibited significant intensity even when tested with sage essential oil. Consequently, the presence of sage can be recognised by an elevated CH band in comparison to the OH band. This observation is evident in Table 7, which displays the spectra of three samples: electrospun PVA nanofibres, the sage essential oil spectrum provided by the supplier, the spectrum of the Solvent Black 3 dye, and the spectrum of the PVA + sage oil + Solvent Black 3 nanofibres. The intensity assessment revealed that the presence of sage led to an augmentation in the intensity of the CH stretching band (I_2915_) in relation to the OH stretching band (I_3300_). As a result, the calculation of the ratio between these two bands (I_3300_/I_2915_) exhibited a substantial contrast between PVA (1.9902) and sage (0.0450). This discrepancy signifies that the presence of sage in sample PVA + sage oil + Solvent Black 3 nanofibres (0.6412) was confirmed through a reduction in this ratio compared to PVA devoid of sage.

Likewise, the distinct characteristics of salvia, namely camphor and thujone, displayed C=O stretching vibrations with a central frequency of 1730 cm^−1^. Additionally, the presence of pinene was identifiable by the -C=C- alkene bond, as evidenced by a peak centred at 1640 cm^−1^ [43,44], which corresponds to typical features of terpenes found in essential oils [45]. In the spectra of the nanofibres (Figure 9b), these specific peaks aligned with the designated F and G peaks of the sage oil spectrum. Upon scrutinising the behaviour of these bands within the sage sample utilised in the study, it was evident that the presence of pinene was notably less prevalent in comparison to camphor and thujone. This was reflected by a distinctive peak at 1730 cm^−1^. The data presented in Table 7 demonstrate that PVA nanofibres with sage led to a reduction in the ratio (I_1640_/I_1734_ = 1.3703) when contrasted with PVA without sage (I_1640_/I_1734_ = 1.8633). Consequently, the intensities of these bands provided further substantiation for the existence of sage within the nanofibres.

To identify the Solvent Black 3 dye inside the nanofibres, the so-called H peak (Figure 9b) was observed at wavelength 1520 cm-1 due to the stretching of the azo groups (N=N) [46]. By studying the ratios in Table 7, it can be seen that with the addition of the SB3 dye, the ratio (I_3300_/I_1520_ = 9.1818) increased with respect to that observed when the nanofibres were produced without dye (I_3300_/I_1520_ = 4.7972), which may be evidence of the presence of dye inside the PVA and sage oil nanofibres.

#### 3.2.5. SEM

SEM characterisation was performed to assess whether there was any difference at the nanoscale according to the electrospinning solution. Figure 10a shows the nanofibres generated from the dissolution of AB194 dye in the PVA solution. It can be seen that the aqueous solvent evaporated correctly during the electrospinning process, as the nanofibres did not have pores on their surfaces. The nanofibres were mostly tubular in cross-section, but there were many elongated beads of similar size along them. Due to the long process time of 90 min, a high deposition of the nanofibres and the creation of different nanofibrous layers on the PES collector substrate were observed.

On the other hand, when the solution prepared from the emulsion of sage oil dyed with SB3 dye in an aqueous solution of PVA was electrospun, a different result was obtained. Figure 10b shows the produced nanofibres, which had more beads in their cross-section; in this case, the beads were also more spherical in shape. Previous studies have shown that the spherical beads contain the sage oil used in the emulsion [34]. In both cases, it is observed that the deposition of the nanofibres was completely random due to the use of a static collector. Table 8 shows the average diameter of the nanofibres and beads.

Bead formation in nanofibre veils can be due to multiple factors. However, it is generally attributed to an unstable Taylor cone due to an inexact balance between the feed flow rate and applied voltage. A lower viscosity favours the appearance of larger-diameter beads. When Acid Black was added to 9% PVA, as shown in Table 2, the viscosity slightly increased from 143.88 cP to 177.36; the flow rate and voltage need to be adjusted as described in Table 1; otherwise, beads appear (Figure 10a). Once the parameters were adapted to the solution’s viscosity, no beads were present in the PVA + Acid Black 194 nanofibres, as can be seen in Figure 10b. The nanofibres showed a diameter of around 87.3 nm, and slight widening was observed, which is attributed to the dyestuff presence in the polymer solution, whose size is 271.04 nm. In contrast, the PVA + sage oil + Solvent Black 3 solution had a higher viscosity, 181.66 cP, so the beads formed in the nanofibres were of a higher diameter, 430.20 cP. It can be seen that the beads showed a spherical shape, which is likely due to the round shapes that keep the oil inside [34]. When the nanofibres were measured, it can be easily appreciated (Table 8) that the fineness decreased considerably when the 9% PVA was electrospun from the oil emulsion (68.94 nm) in comparison to the 9% PVA with dyestuff (87.83). This can be due to the stretching generated by the oil sphere in the polymer in order to be fully coated with PVA.

SEM images of the nanofibrous samples that were subjected to a rubbing test demonstrate the non-existence of nanofibres on the polyester fabric after the test (Figure 11a,c). This also demonstrates the low adhesion of the dyed PVA nanofibres to the collector substrate.

In addition, on the control fabric used to rub the nanofibres, a multitude of small bulges appeared along the polyester fibres and between them (Figure 11b,d). These bulges were not present on the original fabric before the test was carried out.

## 4. Discussion

In the electrospinning technique, the use of different compounds can significantly alter the morphology of the nanofibres. The initial characterisation of the two studied solutions shows how the initial values of the PVA solution can be modified by adding various compounds. Nevertheless, very similar electrospinning process values were established for both processes.

The optical microscopy performed on the polymeric solutions demonstrated the difference between dissolving the AB194 dye in an aqueous solution of PVA, where the obtained image was very similar to that of the PVA solution compared with the image obtained from the emulsion of the RB3 dye dissolved in sage essential oil together with the PVA solution. The image of the emulsion showed the microcapsules generated where the coloured oil was stored due to its darker appearance.

The 5 g/L concentration of dye in the PVA solution was shown to be able to colour the nanofibres and be visible to the naked eye. However, when this amount was added to the sage oil and the emulsion was prepared, the produced nanofibres showed no noticeable colouring.

A quantitative analysis of the colour of the nanofibres by means of diffuse reflectance spectroscopy enabled us to establish the exact colour differences between the samples. The PVA and PVA + sage oil + Solvent Black 3 nanofibres presented brightness values very close to the values obtained by the collector substrate where the nanofibres were deposited, i.e., 94.83 and 94.99 L*, respectively. However, the PVA + Acid Black 194 sample presented a lower brightness value than the rest, 76.39 L*, due to the dark colouring obtained. In addition, the latter sample shifted on the *X*-axis shown in Figure 6a, which is explained by the large colour difference in the sample compared with the standard polyester collector sample.

The nanofibre veils produced from the solution of AB194 in the PVA solution had a greyish colour, as can be seen in the lower-left grid of Figure 6b, closer to the blue colour due to its value of −1.73 b*. In the case of the nanofibrous veils produced from the emulsion of sage oil dyed with the SB3 dye in the PVA solution, the location of the point that represents it in the graph is very close to the representative of the PVA nanofibres and the standard polyester fabric, which reflects the non-existence of surface colour in the generated nanofibres.

Rubbing tests are useful to determine what happens to the dyed nanofibres when the fabric is rubbed. The results on the PVA + Acid Black 194 sample showed that when the fabric with nanofibres was subjected to a rubbing test, the brightness of the fabric increased (91.45 L*). This could be attributed to the removal of the nanofibres from the fabric, as this results in a value very similar to that obtained on the collector fabric (95.29 L*). When analysing the control fabric with which the test was carried out, which initially had a value of 95.29 L* as it was the same fabric, this value dropped to 76.91 L*. This decrease could mean that a large part of the nanofibres that were initially on the sample to be tested were transferred to this control fabric. On the PVA + sage oil + Solvent Black 3 sample, no significant changes were observed after the rubbing test.

The previously assumed transfer of the nanofibres to the control fabric was confirmed by the evaluation of the colour difference shown in Figure 8. The obtained values showed a large decrease in the value of the rubbed PVA + Acid Black 194 sample (4.01 DE*ab) compared with the initial value (19.24 DE*ab), which reflects the disappearance of the nanofibres from the surface. Similarly, the value of the control sample increased to 18.95 DE*ab, which means that it can be categorically stated that almost all the nanofibres were transferred to the fabric with which the rubbing test was carried out. In the case of the nanofibres produced from the emulsion, it is worth noting that the values increased with respect to their original values after the rubbing test. The rubbed fabric reached a value of 1.07 DE*ab, a higher value than the 0.34 DE*ab of the original sample. The value of the control sample was 1.89 DE*ab. Therefore, as with the other solution, greater differences in colour were obtained in the rubbed sample because the nanofibres remained adhered to it.

The colourimetry results after the rubbing test on the nanofibres produced from the PVA + sage oil + Solvent Black 3 emulsion confirm the presence of encapsulated dye inside the nanofibres, which comes out when the polymeric shell from the nanofibre is destroyed. For this reason, the nanofibre veil generated from the emulsion looks very similar to the one produced from the PVA solution, as the SB3 dye remained encapsulated inside the fibre and only PVA was present on the outside.

FTIR characterisation was useful in demonstrating the presence of the Acid Black 194 and Solvent Black 3 dyes, as well as a sage essential oil, in the electrospun nanofibres.

SEM microscopy was able to demonstrate that nanofibres can be generated from both solutions and also provided information regarding the morphology of the fibres, highlighting the appearance of elongated beads along the tubular section of the nanofibres electrospun from PVA and AB194. In this case, an average value of 87.83 nm was obtained for the tubular diameter of the nanofibres and an average diameter of 471.04 nm for the beads. SEM images showed the difference when electrospinning the SB3 sage oil emulsion with PVA. These showed a greater number of beads along the nanofibres with a much more spherical shape. The beads had an average diameter of 430.20 nm, while the nanofibres had an average diameter of 68.94 nm, both values being lower than those obtained from the PVA and AB194 solution.

In agreement with diffuse reflectance spectroscopy, SEM images also showed that the nanofibre veils disappeared in the scrubbed sample and were transferred as aggregates to the test fabric.

## 5. Conclusions

Nanofibres have been extensively studied in terms of their technical aspects; however, the aesthetics of nanofibres have hardly been of interest. In recent decades, nanofibres have become integrated into a multitude of sectors due to their specific characteristics. However, in some of them, such as fashion and biomedicine, the aesthetics of these nanometric structures are beginning to be important.

Conventional textile dyeing can be carried out in multiple ways; however, the dyeing of electrospun nanofibrous structures has not been extensively studied. In this study, the influence of producing electrospun nanofibres from two dyed solutions, one from a dye solution and the other from an emulsion, including oil-soluble dyestuffs, was studied. The results proved the ability to incorporate a dye compound by both processes and its correct subsequent electrospinning process with slight modifications in comparison to the polymer electrospun without dyestuff. Furthermore, although both solutions were initially quite similar in colour, a large difference in colour was evident when the nanofibres were obtained.

The results showed a higher colouring of the nanofibres produced from the PVA+ Acid Black 194 solution. Therefore, we conclude that if a product requires visibly coloured nanofibres, dissolving the dye directly in the PVA solution is the best solution.

On the other hand, the electrospinning of the Solvent Black 3 dye emulsion in the essential oil of sage and PVA showed a non-existent surface colouring due to the fact that the dye remained encapsulated inside the nanofibre, so a completely white surface was obtained on the outside. Subsequently, when this fibre was destroyed, the dye was visible on the surface. This result is ideal for those applications where an initial surface colouring modification is not required, and it is necessary to provide evidence when some stimulus comes into contact with the surface and consequently touches nanofibres and destroys them.

In addition, rubbing tests showed the disappearance of the colour when rubbing the collector substrate where the nanofibres were collected, which is evidence of the weak adhesion between the nanofibres and the polyester fabric. The low adhesion between the collector fabric and the electrospun nanofibres can open up different fields of application, but it can also prevent its application in other fields where a stronger bonding of the two structures is required.

This study aimed to open up new fields of application where aesthetics is important or a touch sensor is necessary, based on the two types of nanofibre colourings studied. These conclusions are limited to 9% PVA solutions and the dyestuff tested. Further studies with different polymers, dyestuffs, and surfaces should be conducted to generalise this behaviour.

## Figures and Tables

**Figure 1 polymers-15-03903-f001:**
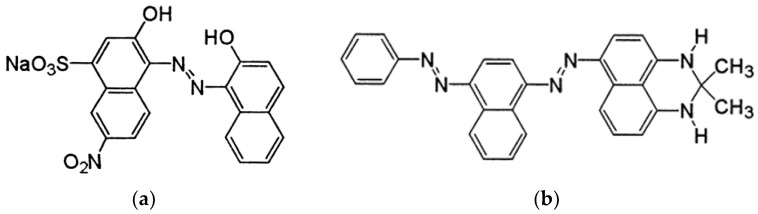
Chemical structures of the dyes. (**a**) Acid Black 194; (**b**) Solvent Black 3.

**Figure 2 polymers-15-03903-f002:**
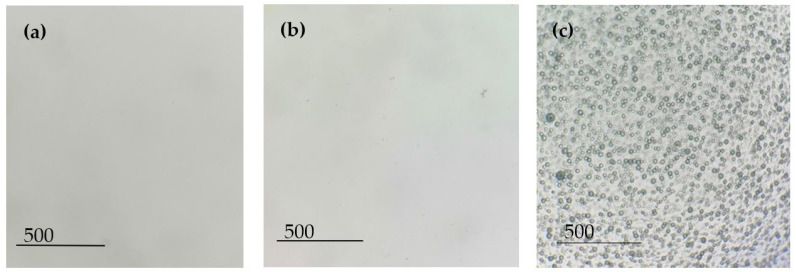
Different solutions under an optical microscope. (**a**) PVA solution; (**b**) PVA + Acid Black 194 solution; (**c**) PVA + sage oil + Solvent Black 3 emulsion.

**Figure 3 polymers-15-03903-f003:**
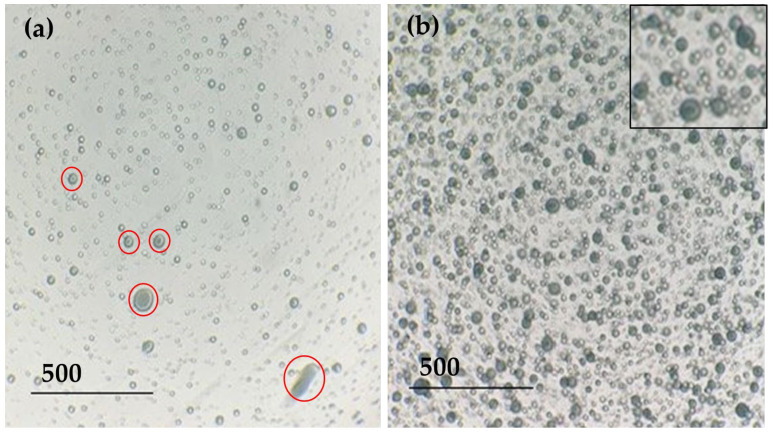
PVA + sage oil + Solvent Black 3 emulsion under an optical microscope. (**a**) Solution extracted from the extruder capillary where the electric current is applied, red circles show different size drops; (**b**) solution extracted from the syringe where the emulsion is held.

**Figure 4 polymers-15-03903-f004:**
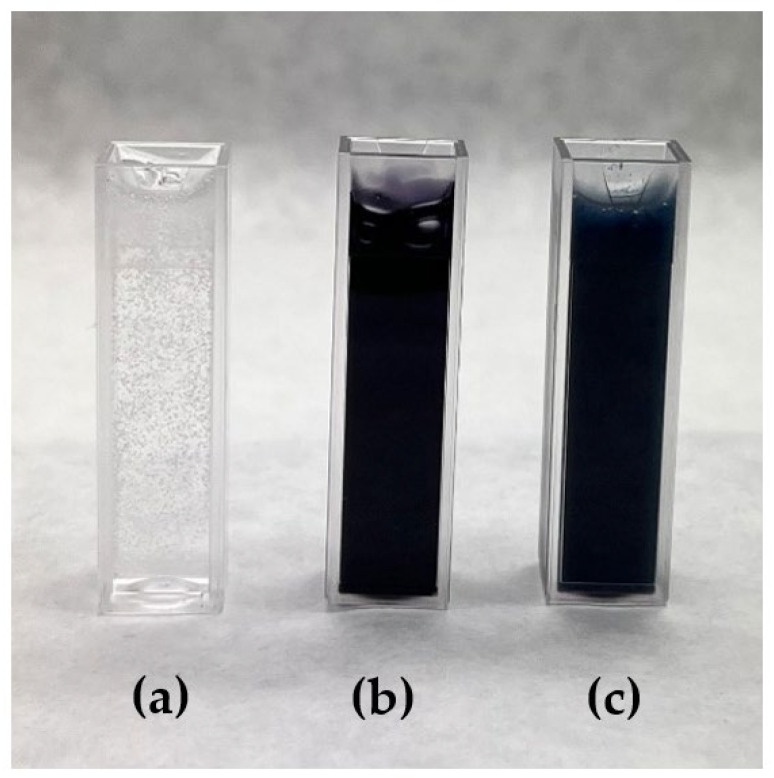
Solutions used in the experiment. (**a**) PVA 9%; (**b**) PVA + Acid Black 194; (**c**) PVA + sage oil + Solvent Black 3.

**Figure 5 polymers-15-03903-f005:**
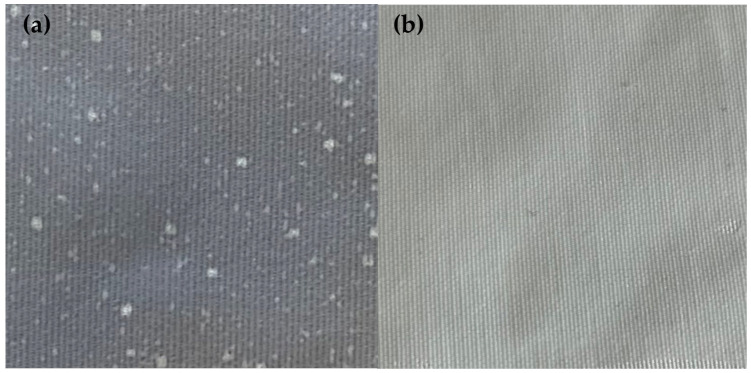
Nanofibres on polyester fabric. (**a**) PVA + Acid Black 194 nanofibres; (**b**) PVA + sage oil + Solvent Black 3 nanofibres.

**Figure 6 polymers-15-03903-f006:**
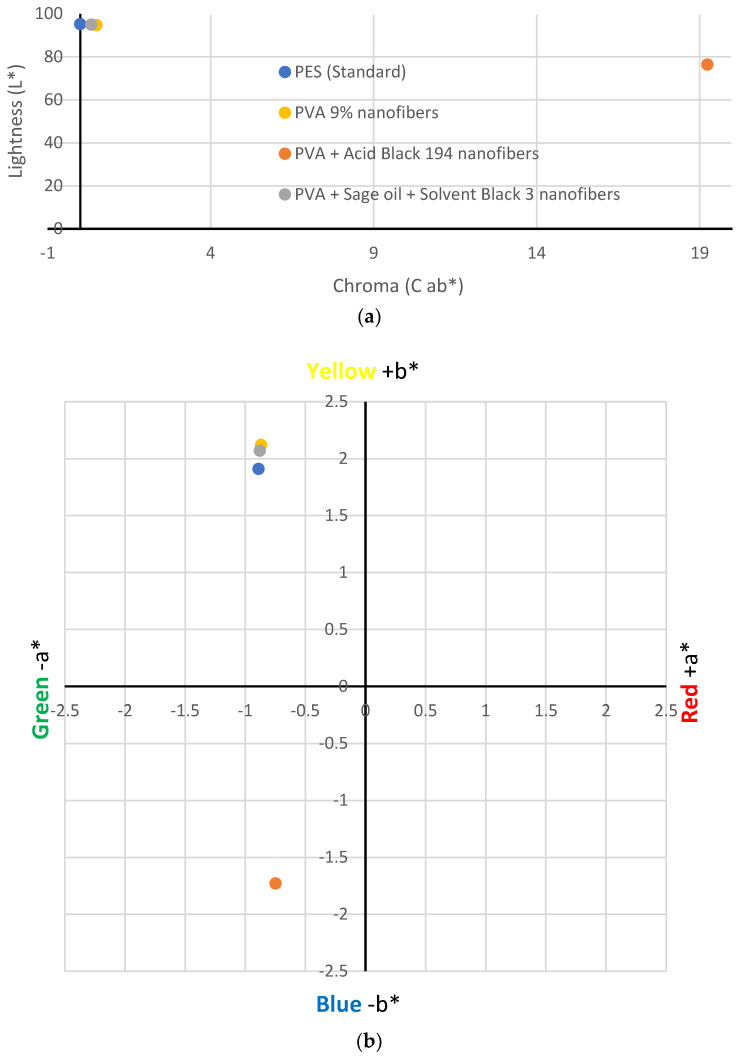
Results of the colourimetry of the samples. (**a**) Sample brightness results. (**b**) Chromatic coordinates: blue point: PES fabric (standard); yellow point: PVA 9% nanofibres; orange point: PVA + Acid Black 194 nanofibres; grey point: PVA + sage oil + Solvent Black 3 nanofibres.

**Figure 7 polymers-15-03903-f007:**
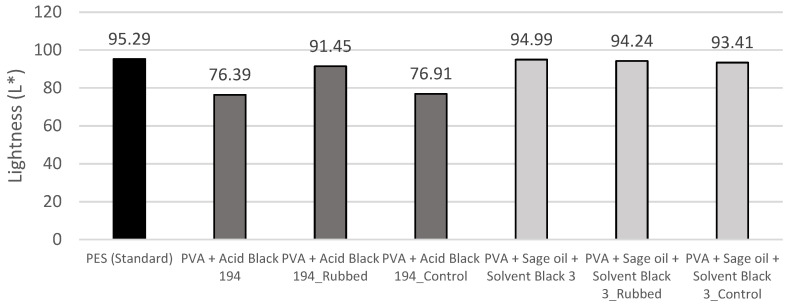
Luminescence (L*) results of the samples after the rubbing test.

**Figure 8 polymers-15-03903-f008:**
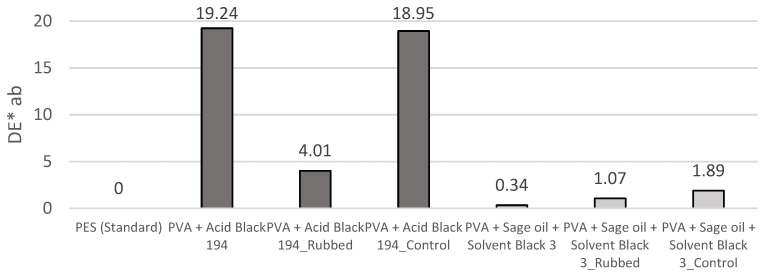
Results of colour difference (DE*ab) of the samples after the rubbing test.

**Figure 9 polymers-15-03903-f009:**
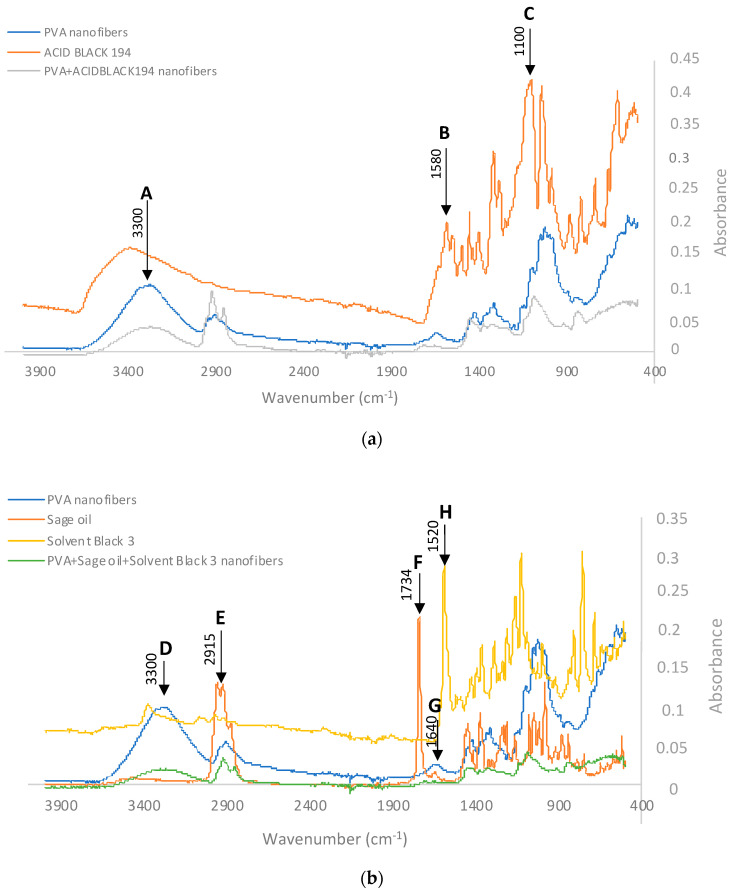
Infrared spectra of the nanofibres. (**a**) Acid Black 194; blue line: PVA nanofibres; orange line: Acid Black 194 dye; grey line: PVA+Acid Black 194 dye nanofibres. (**b**) Solvent Black 3; blue line: PVA nanofibres; orange line: sage oil; yellow line: Solvent Black 3 dye; green line: PVA + sage oil + Solvent Black 3 dye nanofibres.

**Figure 10 polymers-15-03903-f010:**
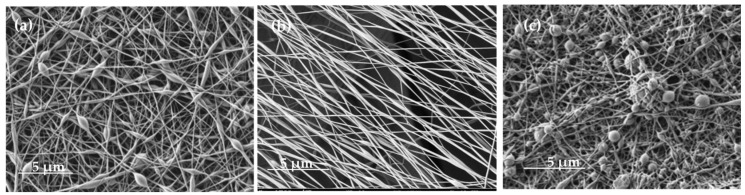
SEM images of the nanofibres. (**a**) PVA + Acid Black 194 nanofibres (5.00 kx) in bad condition (with beds); (**b**) PVA + Acid Black 194 nanofibres (5.00 kx); (**c**) PVA + sage oil + Solvent Black 3 nanofibres (5.00 kx).

**Figure 11 polymers-15-03903-f011:**
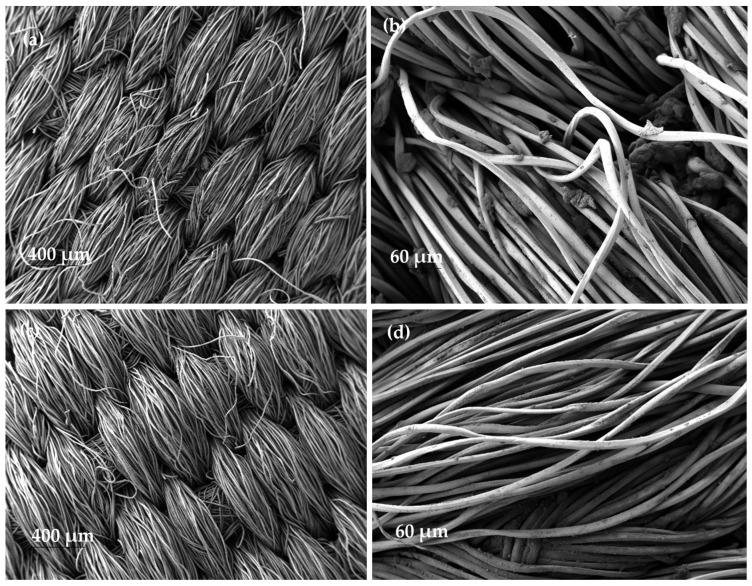
SEM images of the nanofibres after the rubbing test. (**a**) PVA + Acid Black 194 nanofibres (24 ×); (**b**) control fabric for the rubbing test on PVA + sage oil + Solvent Black 3 nanofibre sample (200 ×); (**c**) PVA + sage oil + Solvent Black 3 nanofibre (24 ×); (**d**) control fabric for the rubbing test on PVA + sage oil + Solvent Black 3 nanofibre sample (200 ×).

**Table 1 polymers-15-03903-t001:** The electrospinning parameters.

Reference	PVA	PVA + Acid Black 194	PVA + Sage oil + Solvent Black 3
Flow Rate (mL/h)	0.3	0.45	0.35
Voltage (kV)	10.5	10	11.5
Nozzle–collector distance (cm)	15	15	15
Process time (min)	90	90	90
Room temperature (°C)	24	22.5	26
Room humidity (%)	44	57	40

**Table 2 polymers-15-03903-t002:** Solution parameters.

Reference	PVA 9%	PVA + Acid Black 194	PVA + Sage Oil + Solvent Black 3
Viscosity (cP) (SD 3.47)	143.88	177.36	181.66
Conductivity (µS) (SD 6.78)	294.2	2620	248.8
Surface tension (mN/m) (SD 1.76)	59.58	54.24	33.86
pH (SD 0.03)	5.62	6.32	5.09

**Table 3 polymers-15-03903-t003:** Visual colour test results.

Reference	PVA 9%	PVA + Acid Black 194	PVA + Sage Oil + Solvent Black 3
Colour Identification (five volunteers)	White (5)	Black (5)	White (5)

**Table 4 polymers-15-03903-t004:** CIE Lab* data of the samples.

Reference	L* (SD 0.81)	a* (SD 0.19)	b* (SD 0.21)	dL*	da*	db*	dE*ab	Difference
PES (Standard)	95.29	−0.89	1.91	-	-	-	-	-
PVA 9%	94.83	−0.87	2.12	−0.46	0.02	0.21	0.50	No
PVA + Acid Black 194	76.39	−0.75	−1.73	−18.89	0.14	−3.64	19.24	Yes
PVA + sage oil + Solvent Black 3	94.99	−0.88	2.07	−0.3	0.01	0.16	0.34	No

**Table 5 polymers-15-03903-t005:** CIE Lab* data of the samples after the rubbing test.

Reference	L* (SD 0.76)	a* (SD 0.17)	b* (SD 0.18)	DL*	Da*	Db*	DE*ab	Difference
PES (Standard)	95.29	−0.89	1.91	-	-	-	-	-
PVA + Acid Black 194_Rubbed	91.45	−0.94	0.78	−3.84	−0.05	−1.14	4.01	Yes
PVA + Acid Black 194_Control	76.91	−0.64	−2.7	−18.38	0.25	−4.62	18.95	Yes
PVA + sage oil + Solvent Black 3_Rubbed	94.24	−0.78	2.11	−1.05	0.11	0.19	1.07	Yes
PVA + sage oil + Solvent Black 3_Control	93.41	−0.82	2.06	−1.88	0.07	0.14	1.89	Yes

**Table 6 polymers-15-03903-t006:** Band intensity of FTIR spectra of electrospun PVA nanofibres, Acid Black 194 dye, and electrospun PVA+ Acid Black 194 nanofibres.

Reference	A I_3300_	B I_1580_	B/A I_1580_/I_3300_	C I_1100_	A/C I_3300_/I_1100_
PVA nanofibres	0.3435	0.0772	0.1650	0.0618	0.7946
ACID BLACK 194	0.1591	0.1922	1.2080	0.4142	0.3841
PVA + Acid Black 194 nanofibres	0.0364	0.0492	1.3516	0.0777	0.4684

**Table 7 polymers-15-03903-t007:** Band intensity of the FTIR spectra of electrospun PVA nanofibres, sage oil, and PVA+ sage oil + Solvent Black 3 nanofibres.

Reference	D I_3300_	E I_2915_	D/E I_3300_/I_2915_	F I_1734_	G I_1640_	G/F I_1640_/I_1734_	H I_1520_	D/H I_3300_/I_1520_
PVA nanofibres	0.1017	0.0511	1.9902	0.0139	0.0259	1.8633	0.0212	4.7972
Sage oil	0.0058	0.1288	0.0450	0.2193	0.0143	0.0652	0.0040	1.4500
Solvent Black 3	0.0871	0.0849	1.0259	0.0597	0.0570	0.9547	0.2846	0.3092
PVA + sage oil + Solvent Black 3 nanofibres	0.0202	0.0315	0.6412	0.0027	0.0037	1.3703	0.0022	9.1818

**Table 8 polymers-15-03903-t008:** Average diameter of the nanofibres and beads.

Reference	PVA + Acid Black 194	PVA + Sage Oil + Solvent Black 3
Ø Nanofibres (nm) (SD 2.81)	87.83	68.94
Ø Beads (nm) (SD 1.99)	271.04	430.20

## Data Availability

The data presented in this study is available on request from the corresponding author.

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
