# Peer review of "Emulsion Nanofibres as a Composite for a Textile Touch Sensor"

_polymers, 2023, doi:10.3390/polym15193903_

Round 1

Reviewer 1 Report

The authors conducted a study on Emulsion nanofibres as a composite for a textile touch sensor. The application of nanofibrous veils in multiple industrial sectors has increased dramatically in the last decades. The topic is important and relevant. The introduction provides sufficient background and includes all relevant references. the methodology is correct. The novelty is good. there are some comments that can help improve the manuscript. 

What was the basic reason for selecting the voltage and Nozzle-collector distance? 

Please explain the reason for the beads in Figure 9. 

Please mention the statistical methods.

Please mention the magnification in SEM figures.

Please improve the conclusion section.

What were the limitations of the study?

Minor editing of English language required.

Author Response

Please see the attachment where you can easily identify with blue color our answer and in red one the changes made in the manuscript.

Reviewer 2 Report

The manuscript attempted to make a textile composite composed of a dyed nanofiber layer made of PVA polymers electrospun onto a polyester fabric. Revision is needed. The detailed comments are listed as follows:

Section 2.1. Materials

(a)   This section does not mention any information about the polyester fabric used in this study. Please clearly state the specifications of the polyester fabric, including the fabric type, structure, weight, thickness and density.

(b)   Regarding Acid Black 194 and Solvent Black 3, please provide information about their chemical structure.

Section 2.2. Methods

(a)   Please subdivide this section further into sub-sections, such as preparation of PVA solutions, electrospinning, instrumental characterization, etc.

(b)   Regarding Visco Elite R viscometer, please provide information about the manufacturer and the origin if possible.

(c)   Line 145, concerning the measurement parameters of reflectance spectroscopy, please state the size of the aperture used for measurement.

Section 3.1. Solitions characterization

Typo of this subtitle: should be ‘solution characterization’.

Washing test

Instead of rubbing test, it is suggested that the authors can also conduct washing test to evaluate the colorimetric properties of both types of nanofibers when subjected to a washing process.

Limitations of the study

It is highly recommended that the authors should also discuss the possible limitations of this study so that the manuscript would be more objective and comprehensive.

Author Response

(The authors gave the same response as above.)

Round 2

Reviewer 2 Report

The questions and recommendations are well addressed and the revised manuscrpt is acceptable for publication.